# Programmed Death 1 and Cytotoxic T-Lymphocyte-Associated Protein 4 Gene Expression in Peripheral Blood Mononuclear Cells Can Serve as Prognostic Biomarkers for Hepatocellular Carcinoma

**DOI:** 10.3390/cancers16081493

**Published:** 2024-04-13

**Authors:** Ji Ah Lee, Hei-Gwon Choi, Hyuk Soo Eun, Jiyoon Bu, Tae Min Jang, Jeongdong Lee, Chae Yeon Son, Min Seok Kim, Woo Sun Rou, Seok Hyun Kim, Byung Seok Lee, Ha Neul Kim, Tae Hee Lee, Hong Jae Jeon

**Affiliations:** 1Department of Biological Sciences and Bioengineering, Inha University, 100 Inha-ro, Michuhol-gu, Incheon 22212, Republic of Korea; dlwldk9901@inha.edu (J.A.L.); jbu@inha.ac.kr (J.B.); 2Department of Medical Science, Chungnam National University, 266, Munhwa-ro, Jung-gu, Daejeon 35015, Republic of Korea; hundred4120@naver.com (H.-G.C.); hyuksoo@cnuh.co.kr (H.S.E.); tsb04254@naver.com (H.N.K.); 3Department of Internal Medicine, College of Medicine, Chungnam National University, 266, Munhwa-ro, Jung-gu, Daejeon 35015, Republic of Korea; rws00@cnuh.co.kr (W.S.R.); midoctor@cnu.ac.kr (S.H.K.); gie001@cnuh.co.kr (B.S.L.); 4Department of Internal Medicine, Chungnam National University Hospital, 282, Munhwa-ro, Jung-gu, Daejeon 35015, Republic of Korea; 5Department of Biological Engineering, Inha University, 100 Inha-ro, Michuhol-gu, Incheon 22212, Republic of Korea; 12180838@inha.edu (T.M.J.); chaeyeon1531@inha.edu (C.Y.S.); 6Department of Biomedical Laboratory Science, Daegu Health College, 15 Yeongsong-ro, Buk-gu, Daegu 41453, Republic of Korea; jdjjang95@naver.com (J.L.); msk5615@naver.com (M.S.K.); 7Department of Internal Medicine, Chungnam National University Sejong Hospital, 20, Bodeum 7-ro, Sejong 30099, Republic of Korea

**Keywords:** hepatocellular carcinoma, prognostic biomarkers, programmed death 1, cytotoxic T-lymphocyte-associated protein 4

## Abstract

**Simple Summary:**

This study aimed to investigate the prognostic potential of programmed cell death protein 1 (PD-1) and cytotoxic T-lymphocyte-associated protein 4 (CTLA-4) gene expressions in peripheral blood mononuclear cells for hepatocellular carcinoma. Higher PD-1 gene expression levels were observed in patients with multifocal tumors and those with lymphatic invasion or distant metastasis, while conventional serum biomarkers used for liver function testing did not exhibit similar correlations. PD-1 and CTLA-4 gene expressions demonstrated a strong association with overall survival and progression-free survival in HCC patients, whereas no significant correlation was found among the serum biomarkers used in this study. The findings of this study provide valuable insights into the potential use of PD-1 and CTLA-4 gene expressions as prognostic biomarkers in HCC.

**Abstract:**

Hepatocellular carcinoma (HCC) is a highly aggressive form of liver cancer with poor prognosis. The lack of reliable biomarkers for early detection and accurate diagnosis and prognosis poses a significant challenge to its effective clinical management. In this study, we investigated the diagnostic and prognostic potential of programmed cell death protein 1 (PD-1) and cytotoxic T-lymphocyte-associated protein 4 (CTLA-4) expression in peripheral blood mononuclear cells (PBMCs) in HCC. PD-1 and CTLA-4 gene expression was analyzed comparatively using PBMCs collected from HCC patients and healthy individuals. The results revealed higher PD-1 gene expression levels in patients with multifocal tumors, lymphatic invasion, or distant metastasis than those in their control counterparts. However, conventional serum biomarkers of liver function do not exhibit similar correlations. In conclusion, PD-1 gene expression is associated with OS and PFS and CTLA-4 gene expression is associated with OS, whereas the serum biomarkers analyzed in this study show no significant correlation with survival in HCC. Hence, PD-1 and CTLA-4 expressed in PBMCs are considered potential prognostic biomarkers for patients with HCC that can facilitate prediction of malignancy, response to currently available HCC treatments, and overall survival.

## 1. Introduction

Hepatocellular carcinoma (HCC), a highly hazardous liver cancer, imposes a significant global health burden. It stands as the most common primary malignant disease of the liver and is the leading cause of cancer-related deaths worldwide [1]. Previous reports have demonstrated its aggressive behavior, rapid progression, and limited treatment options. This disease often develops in individuals with underlying liver diseases such as chronic hepatitis B or C, cirrhosis, or nonalcoholic fatty liver disease [2]. HCC tends to be asymptomatic in its early stages, often leading to diagnosis at advanced stages and resulting in a poor prognosis, with a 5-year survival rate of only 10% [3]. The lack of effective therapies contributes to the high mortality rate associated with HCC [4]. There is an urgent need for improved prognostic biomarkers for HCC to enhance patient outcomes, prolong survival, and effectively manage this lethal disease with a rising incidence.

Despite the increasing incidence and impact of HCC, reliable biomarkers for its early detection and accurate diagnosis and prognosis are significantly lacking. Currently, HCC diagnosis primarily relies on imaging techniques, such as ultrasound, computed tomography (CT), magnetic resonance imaging (MRI), and serum biomarkers such as alpha-fetoprotein (AFP) [5]. However, the low sensitivity and specificity of these methods often result in delayed or missed diagnoses, reduced treatment efficacy, and poor patient outcomes. Overall, the absence of effective biomarkers poses a major challenge for clinical management, underscoring the need for novel approaches to detect and prognosticate HCC effectively [6].

Recently, the analysis of gene expression in peripheral blood mononuclear cells (PBMCs), which are any peripheral blood cells having a round nucleus including lymphocytes, monocytes, natural killer cells, and dendritic cells, has emerged as a promising strategy for identifying potential biomarkers of various diseases, including HCC [7]. Programmed cell death protein 1 (PD-1) and cytotoxic T-lymphocyte-associated protein 4 (CTLA-4) are key candidate biomarkers [8]. PD-1 and CTLA-4 are immune checkpoint molecules crucial for regulating T-cell responses and maintaining immune homeostasis [9,10]. Aberrant expression of PD-1 and CTLA-4 has been observed in several types of cancer [11]. Studies have indicated that altered PD-1 and CTLA-4 gene expression in PBMCs correlates with tumor development, progression, and response to therapy [12]. Despite reports suggesting the effectiveness of analyzing immune-related genes in PBMCs as biomarkers for treatment effectiveness against various tumor types, such as squamous cell laryngeal carcinoma and thymoma [13,14], there have been few studies on the effectiveness of these genes as biomarkers for HCC. In particular, immunotherapies targeting PD-1, CTLA-4, and PD-L1, the ligand of PD-1, have recently been developed for HCC treatment and have shown promising effects. While several previous studies have shown that overexpression of PD-L1 in HCC cells is associated with a poor prognosis, liver biopsy to evaluate PD-L1 expression is invasive and carries risks, limiting its clinical use in evaluating PD-L1 expression in HCC patients [15,16,17]. Hence, it is worth investigating whether assessing PD-1 and CTLA-4 using blood samples would be useful in diagnosing or predicting the prognosis of HCC. Although PD-1 or CTLA-4 expression and its significance in the tumor microenvironment have been investigated in HCC patients, most previous studies focused on the isolation of specific immune cells [18].

This study aimed to explore the clinical utility of PD-1 and CTLA-4 gene expression in total PBMCs, without further separation procedures, and investigate the prognostic potential of these markers in HCC. We comparatively assessed PD-1 and CTLA-4 gene expression levels in patients with HCC and healthy individuals to explore their diagnostic and prognostic potential for HCC. Additionally, we investigated the clinical utility of PD-1 and CTLA-4 in determining key pathological tumor characteristics, including tumor size, disease stage, and multifocality. Furthermore, we evaluated the potential use of PD-1 and CTLA-4 expression to predict the survival outcomes of patients with HCC. Our results may indicate the effectiveness of PD-1 and CTLA-4 as biomarkers for HCC and potential immunotherapy targets, enhancing our understanding of immune-related mechanisms underlying HCC and facilitating the development of personalized treatment strategies.

## 2. Materials and Methods

### 2.1. Sample Collection

This study was approved by the Institutional Review Board (IRB) of Chungnam National University Hospital (CNUH), Daejeon, Korea (CNUH-2020-08-049-001 and CNUH-2020-10-088-023). The study cohort included 46 patients, radiologically and pathologically diagnosed with HCC and treated according to standard clinical practice, selected from data registered at the Chungnam National University Hospital Biobank, as well as 10 healthy individuals registered at the Chungbuk National University Hospital Biobank. Blood samples were collected from all participants. Clinical information, including age, modified Union for International Cancer Control (mUICC) stage, metastasis, survival outcome, and serum antigen levels, was provided by the Chungnam National University Hospital Biobank. All participants involved in this study provided signed informed consent before their enrollment.

As illustrated in Appendix A, patients in our study underwent various treatments following standard clinical practices. Notably, 38 patients initially received curative treatments (surgical resection or radiofrequency ablation [RFA]), followed by various subsequent interventions, with the most common being transarterial chemoembolization (TACE). In contrast, eight patients initially underwent treatment with TACE or chemotherapy combined with cyclophosphamide (CTx). The majority of patients received additional treatments, and their tumor burden or clinical status changed throughout the treatment process from diagnosis to death or completion of the study. It is essential to highlight that in South Korea, TACE and curative resections are the most common options for treating patients with HCC, despite the relatively modest 5-year survival rate associated with these [19,20] interventions. Overall survival (OS) was defined as the time from initiation of treatment to death from any cause. Progression-free survival (PFS) is defined as the time from initiation of treatment to disease progression or death from any cause. We received the results of clinical information.

### 2.2. Measurement of Serum Biomarkers

AFP serum concentrations were determined by electrochemiluminescence immunoassay (ECLIA) using the Cobas e801 module (Roche Diagnostics, Basel, Switzerland). Aspartate aminotransferase (AST) and alanine transaminase (ALT) activities were determined using a spectrophotometric method, and serum concentrations of albumin were determined using a bromocresol green colorimetry method. All assays were performed on a Cobas c501 biochemistry analyzer (Roche Diagnostics).

### 2.3. Buffy Coat Separation

All blood samples were centrifuged sequentially at 2990 rpm for 10 min each; then, buffy coats (200 µL) were collected and stored at −80 °C as plasma. The pretreatment process for DNA extraction of sample is as follows: The 200 μL of buffy coat was treated with 20 μL proteinase K to remove erythrocyte and ascites. The reaction ended, buffy coat and ascites were mixed with 200 μL lysis buffer by pulse-vortexing for 15 s, then incubated at 57 °C for 15 min. The sample was added to 200 μL 95% ethyl alcohol (Samchun, Seoul, Korea) and mixed again by pulse-vortexing for 15 s. The bead was reacted with the pretreated sample for 10 min. The bead was carefully dipped into 350 μL buffer AW1 (Qiagen, Hilden, Germany) for 60 s, then dipped into 50 μL Rnase/Dnase free water for 3 min. The elution was stored at −80 °C. DNA was extracted using a QIAamp DNA Mini kit (Qiagen) following the instructions provided by the manufacturer as described previously [21,22].

### 2.4. Sample Processing

Gene amplification was performed using a Rotor-Gene 6200 real-time cycler (Corbett, Pittsburgh, PA, USA) following a previously reported method [23]. Quantitative polymerase chain reaction (qPCR) was performed on each sample in triplicate, with a final volume of 25 µL for each reaction mix, using a QuantiNova SYBR Green PCR kit (Qiagen). The PCR program included an initial step of 95 °C for 5 min, followed by 40 cycles of 95 °C for 5 s and 60 °C for 10 s.

The following primers were used in this study: PD-1 gene (Hs00793019_CE, Invitrogen, Waltham, MA, USA), CTLA-4 gene (Hs00753634_CE, Invitrogen), RPP30 forward primer (5′-GATTTGGACCTGCGAGCG-3′), and RPP30 reverse primer (5′-GCGGCTGTCTCCACAAGT-3′). The expression levels of PD-1 and CTLA-4 genes were comparatively analyzed using the 2^−ΔCt^ method, where ΔCt = Ct (PD-1 or CTLA-4) − Ct (RPP30), with RPP30 used as the reference gene [23,24].

### 2.5. Statistical Analysis

Serum levels of biomarkers and 2^−ΔCt^ DNA expression levels were compared between the subgroups using either Student’s *t*-test or Mann–Whitney U test. Specifically, continuous data (e.g., gene expression and serum antigen expression) were compared using Student’s *t*-test (unpaired two-sample *t*-test) in cases where the sample distribution was normal, and the nonparametric Mann–Whitney U test was used to compare two groups when the sample distribution exhibited asymmetry. To evaluate clinical capabilities, a receiver operating characteristic (ROC) curve was analyzed. Survival was evaluated using Kaplan–Meier plots. A Cox regression model was used to assess OS and PFS. Statistical significance was set at *p* < 0.05. All statistical analyses were performed using SPSS Statistics (version 20.0; IBM Corp., Armonk, NY, USA).

## 3. Results

### 3.1. Baseline Characteristics of Study Subjects

Most (33 patients, 71.7%) of the patients with HCC underwent RFA as their primary treatment modality. Five patients underwent surgical resection, TACE was administered to four, and sorafenib therapy was administered to four patients. Of the 33 patients who underwent RFA as the initial treatment, the disease recurred in 22, and these patients received a second treatment during the observation period. TACE was the most common (applied to 13 patients) second treatment modality, whereas RFA was administered as the second treatment in 9 patients. A third treatment for disease progression or recurrence was applied to 18 of the 33 patients who received RFA as the initial treatment; among them, TACE, RFA, and chemotherapy were applied to 10, 6, and 2 patients, respectively. Of the patients who received surgical resection as the initial treatment, two patients underwent surgical resection and two patients received TACE as a second treatment. Of the four patients who received TACE as the initial treatment, two received a second treatment, one receiving RFA and the other receiving TACE. Treatments for the enrolled patients are summarized in Appendix A.

Blood samples were collected from all participants before treatment initiation. Clinical data, including pathological stage, tumor multiplicity, lymph node involvement, and presence of metastasis, were obtained using CT imaging and/or biopsy reports. Detailed patient characteristics are summarized in Table 1.

### 3.2. The Diagnostic Capability of PD-1 and CTLA-4 Gene Expression for the Diagnosis of Hepatocellular Carcinoma

The expression of PD-1 and CTLA-4 in PBMCs of patients with HCC and healthy donors was compared (Figure 1). In the patients with HCC, the average Ct values for PD-1 and CTLA-4 genes, normalized to RRP30, were 1.24 ± 1.05 and 1.17 ± 1.17, respectively (Figure 1A). These values were higher than those obtained from healthy donors (PD-1: 1.16 ± 0.55, *p* = 0.814; CTLA-4: 1.08 ± 0.50, *p* = 0.832); however, these differences were not statistically significant. The areas under the curve (AUC) of the ROC were 0.428 (*p* = 0.391) and 0.452 (*p* = 0.606) for PD-1 and CTLA-4, respectively (Figure 1B). These associations were not significant, while the conventional serum marker AFP exhibited significantly elevated expression in HCC patients (286.57 ± 1768.99 vs. 1.25 ± 1.57) with an AUC-ROC of 0.957 (*p* < 0.001) compared to that in healthy individuals. Other standard liver function serum markers, except for AST, showed no association with tumor presence (Figure 1 and Appendix A).

### 3.3. The Utility of PD-1 and CTLA-4 Gene Expression as Biomarkers for the Pathological Characteristics of Hepatocellular Carcinoma

We subsequently investigated the utility of PD-1 and CTLA-4 as biomarkers for pathological characteristics, including stage, size, multifocality, nodal status, and presence of distant metastasis, of tumors (Figure 2A). A tendency of positive correlation between PD-1 gene expression and tumor multifocality was detected. Patients with multifocal tumors tended to exhibit higher PD-1 gene expression levels than those with single tumors (1.504 ± 1.070 vs. 0.973 ± 0.803; *p* = 0.067). There was no difference in CTLA-4 gene expression between patients with multifocal tumors and those with single tumors (1.252 ± 0.904 vs. 0.911 ± 0.654; *p* = 0.152). No serum antigen biomarker exhibited a correlation with multifocal tumors (*p* > 0.400).

Meanwhile, PD-1 expression was associated with lymphatic invasion and distant metastasis (Figure 2B). PD-1 expression was elevated in patients with nodal invasion or distant metastasis compared to that in patients without these features (1.075 ± 0.901 vs. 0.239 ± 0.767; *p* = 0.044). However, no statistically significant difference was detected in CTLA-4 expression between patients with distant metastases or nodal invasion and those without these features. Considering the serum antigens analyzed in this study showed no association with nodal invasion or distant metastasis, PD-1 may be a potential candidate indicator of high malignancy in HCC. However, the serum biomarkers exhibited no significant correlation with tumor size and mUICC stage (Appendix A).

### 3.4. The Prognostic Predictability of PD-1 and CTLA-4 Gene Expression in Patients with Hepatocellular Carcinoma

The prognostic potential of PD-1 and CTLA-4 gene expression for predicting survival outcomes was rigorously evaluated using statistical techniques, including Kaplan–Meier survival analysis and Cox proportional hazard regression modeling (Figure 3 and Figure 4). In the study cohort, 14 patients died, while 34 others experienced disease progression during the study period. Our results revealed a significant correlation between PD-1 expression and OS in patients with HCC (Figure 3A). Notably, patients with PD-1 expression levels above the median value showed a mean OS of 36.030 ± 9.005 months, whereas those with PD-1 expression levels below the median exhibited no mortality events (*p* < 0.001). Similarly, CTLA-4 expression was closely associated with patient survival; patients showing CTLA-4 expression levels above the median value showed a mean OS of 83.426 ± 13.992 months compared to 125.828 ± 12.455 months for those with CTLA-4 expression levels below the median (*p* = 0.014). Furthermore, Kaplan–Meier analysis indicated a consistent correlation between PD-1 gene expression and survival rate, even when the patient cohort was stratified by quartiles of gene expression (Figure 3A, Appendix A).

Similarly, poorer PFS was detected in patients with higher PD-1 expression than in those with lower expression levels (Figure 3B). Specifically, patients with PD-1 expression levels above the median value displayed a notably shorter PFS (22.384 ± 5.953 months) than those with PD-1 expression levels below the median (69.318 ± 14.378 months) (*p* = 0.011). In contrast, patients with higher and lower than average levels of CTLA-4 expression demonstrated mean PFS times of 36.063 ± 7.923 months and 50.841 ± 13.335 months, respectively (*p* = 0.845); hence, CTLA-4 was not an indicator of tumor progression. Notably, no serum markers investigated in this study were significantly correlated with OS or PFS, highlighting the unique prognostic potential of using PD-1 and CTLA-4 gene expression as biomarkers in HCC. However, analyses of ALT and albumin levels revealed that patients with high serum antigen levels had longer OS and PFS than those with low ALT and albumin levels (Figure 3A,B).

Further detailed investigations of the prognostic potential of these genes were conducted. Patients were stratified into two groups based on tumor size (large and small tumor groups) using median values as thresholds. The results indicated that PD-1 expression in PBMCs is an effective biomarker for predicting OS in both groups. Patients with PD-1 expression levels below the median showed no deaths, irrespective of tumor size (*p* = 0.021 in the small-tumor group and *p* = 0.020 in the large-tumor group). We also found that higher PD-1 expression was significantly associated with poorer OS in patients with lower-stage-tumors (mUICC stage 1 or 2, *p* < 0.002). A similar tendency was identified in patients with high-mUICC-stage tumors; however, a correlation could not be established due to an insufficient number of patients (n = 8, *p* = 0.171). In contrast, neither CTLA-4 nor the conventional serum biomarker AFP exhibited similar efficacy in predicting OS in HCC (Appendix A).

Univariate Cox regression analysis, used to evaluate the prognostic value of PD-1 and CTLA-4 expression levels, revealed significant associations between PD-1 expression (non-categorical model) in PBMCs and both OS and PFS (hazard ratio [HR] of 2.232, *p* < 0.001 and HR of 1.259, *p* = 0.101, respectively). Although CTLA-4 exhibited a slightly lower prognostic ability than PD-1, it still displayed a significant correlation with OS, with an HR of 1.321 (*p* = 0.042). However, PD-1 expression showed limited effectiveness in predicting tumor progression (HR = 0.986, *p* = 0.917). Considering the median value as a cutoff, a binary model of PD-1 expression revealed HRs of 89.441 (*p* = 0.044) for OS and 2.385 (*p* = 0.015) for PFS. For the binary CTLA-4 model, the HRs for OS and PFS were 4.350 (*p* = 0.025) and 1.086 (*p* = 0.810), respectively. Notably, under various other threshold conditions, PD-1 expression demonstrated a strong correlation with OS. Both PD-1 and CTLA-4 demonstrated superior prognostic abilities over the other serum biomarkers investigated in this study. And this correlation remained similar after adjusting treatment conditions (Appendix A). Collectively, these findings suggest that, while PD-1 and CTLA-4 gene expression levels may not serve as reliable diagnostic markers for HCC, they may serve as prognostic biomarkers for survival outcomes and highly malignant HCC subtypes.

## 4. Discussion

Conventional treatment strategies often fall short in managing HCC, primarily due to the resistance of HCC to traditional therapies [25]. HCC frequently presents at advanced stages with aggressive tumor growth, vascular invasion, and extrahepatic metastasis, making curative surgical resection or transplantation feasible only for a subset of patients. Additionally, the limited efficacy of previous systemic chemotherapy in HCC was largely attributed to the inherent chemoresistance of these tumor cells [26,27].

Meanwhile, various immunotherapies have recently emerged for HCC treatment, with a focus on immune checkpoints. Atezolizumab, a representative anti-PD-1L antibody, has shown superior survival outcomes compared to sorafenib when combined with bevacizumab, becoming the treatment of choice for advanced HCC [28]. The effectiveness of immune checkpoint inhibitors for HCC is attributed to the tumor’s immune evasion mechanisms. Key pathways involved in immune evasion, such as the PD-1 and CTLA-4 pathways, are targeted by most immunotherapeutics used clinically for HCC. These pathways downregulate T-cell activation to maintain peripheral tolerance, creating an immunosuppressive state that allows tumor growth without elimination. While studies have linked PD-L1 overexpression to poorer prognosis, no study has demonstrated an association between PD-1 or CTLA-4 gene expression in PBMCs and survival outcomes in HCC patients [15,16,17].

Our results suggest that PD-1 and CTLA-4 expression may not serve as reliable diagnostic markers for HCC. In our study, the average Ct values of PD-1 and CTLA-4 gene expression in the HCC patient group were slightly higher than those in the healthy donor group; however, these results were not statistically significant. However, this lack of significance may be due to the limited number of cases. Enrolling a larger number of healthy donor cases and conducting future experimental investigations could address this limitation. However, when analyzing the patient data in more detail, a strong correlation between the expression of these immune checkpoint molecules and patient survival was detected. Although the patients included in this study were treated using conventional therapeutic methods, their OS and PFS were strongly associated with expression of the identified immune checkpoint inhibitors, specifically PD-1. Overexpression of PD-1 and CTLA-4 has been linked to an immunosuppressive microenvironment. These immune checkpoint molecules play crucial roles in downregulating immune responses. In tumor cells, these pathways are exploited for immune evasion. The strong correlation between high PD-1 and CTLA-4 expression levels and poor patient outcomes suggests that the ability of tumors to evade the immune system is critical for HCC progression. Specifically, our findings revealed that patients with high PD-1 gene expression at baseline exhibited poor survival outcomes. For future studies, we plan to delve deeper into the prognostic potential of PD-1 genes with various HCC treatment regimens.

This observation aligns with increasing evidence of the effectiveness of immune checkpoint inhibitors that target PD-1 and CTLA-4 as an HCC treatment [29,30]. In future studies, we will analyze PD-1 and CTLA-4 gene expression in PBMCs alongside soluble proteins encoded by these genes in a large cohort of patients with HCC following immune checkpoint inhibitor treatment. As demonstrated in this study, conventional serum biomarkers (e.g., AFP, AST, and ALT) are suboptimal for estimating the survival of HCC patients. Exploring more relevant serum proteins, such as sPD-1, sCTLA-4, sPD-L1, sTIM-3, sBTLA, sVISTA, and sLAG-3, in addition to examining their gene expression levels in PBMCs, may help identify effective serum markers for predicting the survival outcomes of HCC patients [31,32]. Previously, we quantified mRNA and protein expression and established a multimodal analysis platform [22]. The newly developed multimodal liquid biopsy assay was applied to patients and compared with conventional tissue biopsies based on tissue immunostaining.

While studies indicate increased expression of inhibitory receptors, such as PD-1 and CTLA-4 on tumor-infiltrating lymphocytes in HCC tissues, as well as their association with patient prognosis [33,34], the challenge of obtaining biopsy samples from all patients with HCC hampers studies on immune cells associated with cancer lesions. Additionally, expression levels of these immune checkpoint molecules in HCC tumor tissues may differ from those in the peripheral blood of patients with HCC, given that interactions between immune cells and tumor tissues are influenced by multiple molecules in the tumor microenvironment [35]. In light of these challenges, analyzing PD-1 and CTLA-4 gene expression levels in PBMCs, which are relatively accessible, may provide new insights into patients’ prognosis based on the expression of immune checkpoint molecules. Furthermore, comparing PD-1 or CTLA-4 expression between immune cells bound to tissues and in PBMCs may help address scientific questions regarding how these molecules affect tumor progression.

## 5. Conclusions

In conclusion, we demonstrated the potential of PD-1 and CTLA-4 expression in PBMCs to serve as prognostic biomarkers for HCC. Although the results were not statistically significant when compared to those from healthy donors, PD-1 expression was found to be associated with high malignancy. Particularly, it was associated with lymphatic invasion or distant metastasis. Moreover, the expression level of PD-1 was associated with OS and PFS and expression level of CTLA-4 was associated with OS. Thus, the potential of these new prognostic biomarkers for survival in patients with HCC is evident. These findings highlight the importance of further research on the molecular mechanisms underlying PD-1 and CTLA-4 dysregulation, as well as the potential role for these molecules in targeted therapies and personalized treatment strategies for HCC.

## Figures and Tables

**Figure 1 cancers-16-01493-f001:**
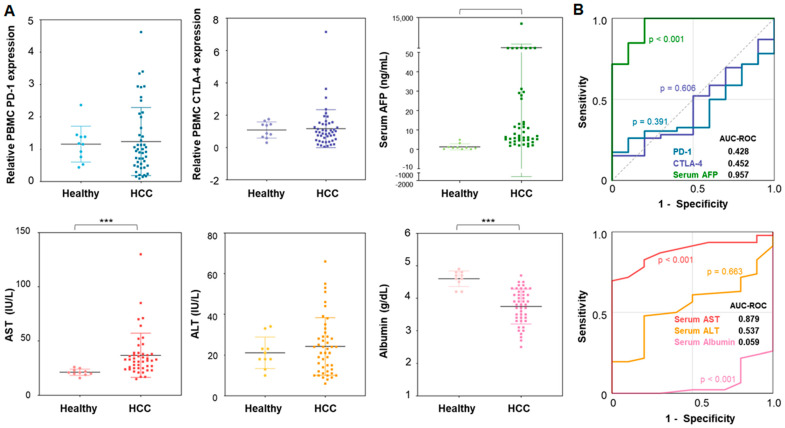
Diagnostic capability of PD-1 and CTLA-4 genes in PBMCs for the detection of HCC compared to the serum AFP level: (**A**) PD-1 gene expressions, CTLA-4 gene expressions, and serum AFP levels between HCC patients and healthy donors; (**B**) the ROC analysis of gene expressions and serum AFP level for discriminating HCC patients from healthy individuals. Note that the significance levels are indicated as *** *p* < 0.010.

**Figure 2 cancers-16-01493-f002:**
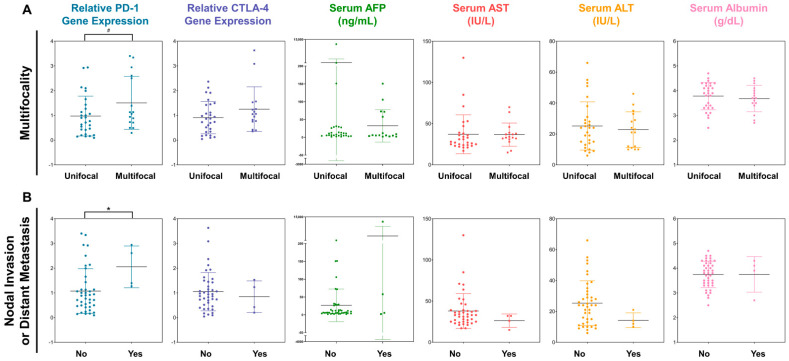
The expression profiles of PD-1 and CTLA-4 genes in PBMCs and serum antigens depending on (**A**) the tumor multifocality and (**B**) the existence of nodal invasion or distant metastasis. Note that the significance levels are indicated as ^#^ *p* < 0.100, and * *p* < 0.050.

**Figure 3 cancers-16-01493-f003:**
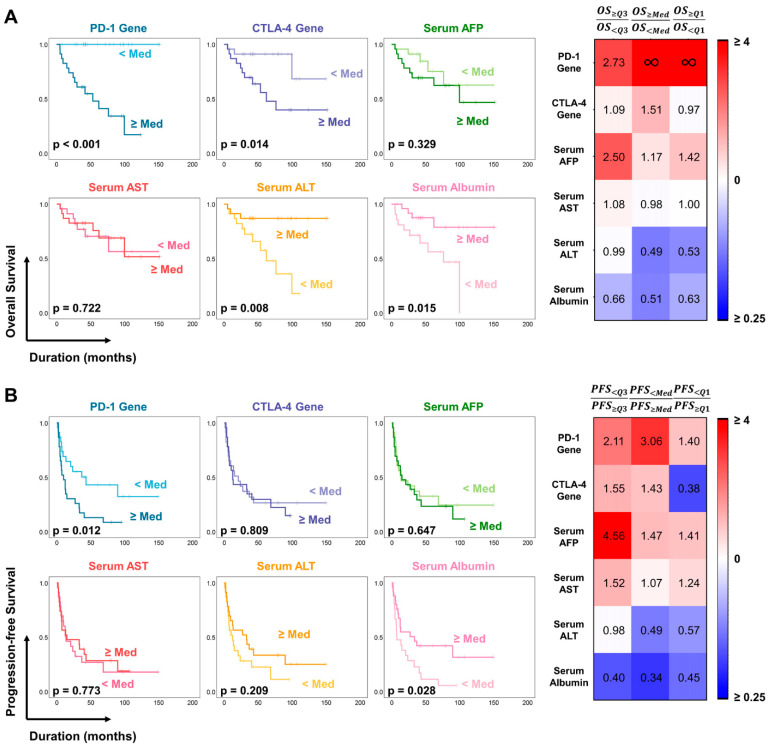
The Kaplan–Meier method for estimating the (**A**) OS and (**B**) PFS of patients with HCC using PD-1 and CTLA-4 gene expressions in PBMC. The results were compared with the standard serum tests for liver function, including AFP, AST, ALT, and albumin.

**Figure 4 cancers-16-01493-f004:**
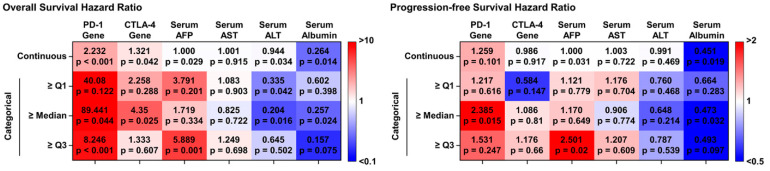
The univariate Cox regression analysis of the immune checkpoint genes and serum biomarkers for estimating OS and PFS.

**Table 1 cancers-16-01493-t001:** Basic characteristics of the recruited HCC patients and healthy volunteers.

	Patients with Hepatocellular Carcinoma (n = 46)	
**The range of dates of sample collection**	From June 2010 to November 2021	
**Age at blood draw**		
Median	64	
Range	39–92	
**The Largest Tumor Size**		
Median	1.6 cm	
Range	1.0–9.0 cm	
Average	1.9 ± 1.2 cm	
	No.	Ratio
**mUICC Stage**		
1	23	50.00%
2	15	32.60%
3	3	6.50%
4	5	10.90%
**Nodal invasion**	2	4.30%
**Metastasis**	2	4.30%
**Multifocality**		
Unifocal Tumor	29	63.00%
Multifocal Tumor	16	34.80%
Undetermined	1	2.20%
	Patients with healthy volunteers (n = 10)	
**The range of dates of sample collection**	From September 2020 to November 2020	
**Age at blood draw**		
Median	26	
Range	19–61	

## Data Availability

The data generated in the present study may be requested from the corresponding authors.

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
