# Peer review of "Programmed Death 1 and Cytotoxic T-Lymphocyte-Associated Protein 4 Gene Expression in Peripheral Blood Mononuclear Cells Can Serve as Prognostic Biomarkers for Hepatocellular Carcinoma"

_cancers, 2024, doi:10.3390/cancers16081493_

Round 1

Reviewer 1 Report

Comments and Suggestions for Authors

The authors investigate the clinical significance of  the expression of PD-1 and CTLA4 in PBMCs in patients with hepatocellular carcinoma. I have some concerns:

The limited sample size is a major shortcoming of the manuscript. As classical and well recognized immune checkpoint molecules, the expression, function and mechanism of

 PD-1 and CTLA-4 has been widely discussed and known, while the author only analyzed the relationship between their expression in PBMCs and disease progression, so the significance and novelty is limited. In addition, the author should follow the rules of scientific writing, and when illurstrating the results, sub-title should be involved to make the results and logic more clear, and fonts in the figure should be identical.

Comments on the Quality of English Language

Minor editing needed.

Author Response

Response : We appreciate the reviewer’s comment. As mentioned by the reviewer, the expression, function and mechanism of PD-1 and CTLA-4 has been widely discussed and known. However, while overexpression of PD-L1 is known as poor prognostic factor in patients with HCC, the prognostic value of PD-1 and CTLA-4 has not been well evaluated. Because evaluation of PD-1L expression requires an invasive liver biopsy, whereas evaluation of PD-1 and CTLA-4 expression can be done using a blood sample, we think that it is worth investigating whether assessing PD-1 and CTLA-4 expression would be useful in diagnosing or predicting prognosis of HCC. Moreover, we believe that our study is valuable because it is the first study to show that gene expression of PD-1 or CTLA-4 in PBMCs is associated with poor survival outcome in patients with HCC. But as the reviewer mentioned, the limited sample size is considered a shortcoming of this study, and large-scale, multicenter studies are needed to overcome these limitations in the future. Finally, we have added sub-title to the Results part and have corrected that fonts in the figure are identical.

Reviewer 2 Report

Comments and Suggestions for Authors

The manuscript titled “Programmed death 1 and cytotoxic T-lymphocyte-associated protein 4 gene expression in peripheral blood mononuclear cells can serve as prognostic biomarkers for hepatocellular carcinoma" accessed expression levels of PD-1 and CTLA-4 genes are claimed its strongly associated with overall survival and progression-free survival. Authors however found that the serum biomarkers analyzed in this study show no significant correlation with survival in HCC. The manuscript is interesting. However, the manuscript requires a significant attention specifically to improve punctuations, grammar and the readability. Also, the introduction and discussion sections currently lack sufficient detail, necessitating the inclusion of additional information to bolster their informativeness. To rectify this deficiency and align with the groundbreaking findings presented in the manuscript, the authors are strongly encouraged to incorporate relevant literature to improvise overall readability of manuscript.

Comments on the Quality of English Language

Please see comment above.

Author Response

Response : We appreciate for the valuable suggestion. As recommended by the reviewer, the English language was edited to improve punctuation, grammar, and readability. And we incorporate relevant literature and added the following to the Introduction part and the Discussion part:

Introduction

In particular, immunotherapies targeting PD-1, CTLA-4, and PD-L1, the ligand of PD-1, have recently been developed for HCC treatment and have shown promising effects. While several previous studies have shown that overexpression of PD-L1 in HCC cells is associated with a poor prognosis liver biopsy to evaluate PD-L1 expression is invasive and carries risks, limiting its clinical use in evaluating PD-L1 expression in HCC patients. Hence, it is worth investigating whether assessing PD-1 and CTLA-4 using blood samples would be useful in diagnosing or predicting the prognosis of HCC.

Discussion

Meanwhile, various immunotherapies have recently emerged for HCC treatment, with a focus on immune checkpoints. Atezolizumab, a representative anti-PD-1L antibody, has shown superior survival outcomes compared to sorafenib when combined with bevacizumab, becoming the treatment of choice for advanced HCC [source]. The effectiveness of immune checkpoint inhibitors for HCC is attributed to the tumor's immune evasion mechanisms. Key pathways involved in immune evasion, such as the PD-1 and CTLA-4 pathways, are targeted by most immunotherapeutics used clinically for HCC. These pathways downregulate T-cell activation to maintain peripheral tolerance, creating an immunosuppressive state that allows tumor growth without elimination. While studies have linked PD-L1 overexpression to poorer prognosis, no study has demonstrated an association between PD-1 or CTLA-4 gene expression in PBMCs and survival outcomes in HCC patients.

Reviewer 3 Report

Comments and Suggestions for Authors

This research analyzed PD1 and CTLA4 in patients with hepatocellular carcinoma, analyzed in peripheral blood mononuclear cells, which means B and T lymphocytes, monocytes, and dendritic cells. What cells expressed those markers is unknown in this study. As a matter of fact, this is very relevant for these two immuno-oncology markers. Knowing this would have improved/level up the quality of this publication. But at this stage of research, this may not be possible. After reading the manuscript, it looks like the authors are "pushing" the statistics to identify significant p values. I would not recommend it. If it is not significant, better leave it as negative result rather than "it was different but not significant....". However, the data of this study may be of interest to the readers.

Additional comments:

(1) Line 74. Regarding PBMCs. I know it is a basic definition. But it may help the readers to remember that a peripheral blood mononuclear cell is any peripheral blood cell having a round nucleus. These cells consist of lymphocytes and monocytes, whereas erythrocytes and platelets have no nuclei, and granulocytes have multi-lobed nuclei. In humans, lymphocytes make up the majority of the PBMC population, followed by monocytes, and only a small percentage of dendritic cells.

(2) Line 90. Could you please describe the meaning of "without further separation procedures". Does it mean the different subtypes of PBMCs are not defined/analyzed?

(3) The clinicopathological characteristics of the patients that are shown in Figure S1 could be included in the main text, in the form of table because it is important information.

(4) Could you please confirm that RPP30 expression level is unaffected by experimental factors? Also, it should show minimal variability in its expression between tissues and physiological states of the organism.

(5) Line 158. Could you please explain why ROC curve was used?

(6) Please add catalog number of all reagents that were used and are described in the material and methods section.

(7) Please define how OS and PFS were calculated. The criteria.

(8) Lines 164-181. How heterogeneous were the treatment of the patients? What clinicopathological characteristics/variables were associated with the prognosis of the patients? Were the "conventional" clinical/pathological variables of HCC analyzed to assess prognosis?

(9) Lines 182-193. As I understand, in the analysis of PD1 and CTLA4 in PBMCs, no differences were found between HCC and controls, is this right? But other "classical" variables such as AST, ALT, and Albumin did show differences (this is why are used in biochemical analyses).

(10) Line 202. Regarding "A strong correlation between PD-1 gene expression and tumor multifocality was detected: patients with multifocal tumors exhibited higher PD-1 gene expression levels than those in other patients (1.504 ± 1.070 vs. 0.973 204 ± 0.803; p = 0.067)". By using strong you seems to imply it is significant, but it is not as it is 0.067. This may mislead the reader.

(11) Line 205. Regarding "CTLA-4 was also overexpressed in patients with multifocal tumors compared to that in other patients; however, the results were not significant (1.252 ± 0.904 206 vs. 0.911 ± 0.654; p = 0.152)". If it is not significant, then better leave it as non different because the p value of 0.05 was decided at the beginning of the experimental design, wasn't it?

(12) Line 208. Regarding "Considering the previously reported association between poor prognosis of HCC with multifocal tumors and the challenges in analyzing multifocality with conventional tissue biopsy methods, PD-1 gene expression could be applied as a useful biomarker for detecting patients with multifocal tumors". I am sorry, but, your data shown no difference, so PD-1 in PBMCs is negative.

(13) Figure 2B. The number of cases in nodal invasion/M1 is very small, only 4? Please confirm. Therefore, the statement of line 219 "PD-1 is proposed as an indicator of high malignancy in HCC" may not be adequate. Please discuss.

(14) Line 348, conclusion, "PD-1 expression was found to be strongly associated with high malignancy.". Please indicate where this data is shown in the manuscript.

(15) Lline 350. "Significant correlation between PD-1 and CTLA-4 gene expression and OS and PFS in patients with HCC was identified". I may be wrong, but yes for PD1 in OS and PFS, but CTLA not in PFS. Please correct conclusion, and abstract.

Reviewer 4 Report

Comments and Suggestions for Authors

In this manuscript, the authors have investigated the diagnostic and prognostic potential of PD-1 and CTLA-4 expression in peripheral blood mononuclear cells (PBMCs) of HCC. Based on the higher expression levels of PD-1 and CTLA-4 in PBMCs and their association with pathological features and survival, the authors had concluded that both the molecules can be used as potential prognostic biomarkers for patients with HCC. The study has clinical significance. However, the following points should be addressed before considering for publication.

1. The number of patients should be increased and statistically justified.

2. How the different treatment conditions had an impact on the levels of PD-1 and CTLA-4 as well their prognostic potential? 

3. The protein levels of PD-1 and CTLA-4 will strengthen the findings.

Comments on the Quality of English Language

Moderate English language editing will improve the manuscript.

Round 2

Reviewer 4 Report

Comments and Suggestions for Authors

The manuscript was improved by the revisions and can be accepted in its current form.

Comments on the Quality of English Language

Moderate English language editing will improve the readability.